# Characteristics and Genomic Localization of Nurse Shark (*Ginglymostoma cirratum*) IgNAR

**DOI:** 10.3390/ijms252312879

**Published:** 2024-11-29

**Authors:** Wenjie Tang, Kaixi Zheng, Shengjie Sun, Bo Zhong, Zhan Luo, Junjie Yang, Lei Jia, Lan Yang, Wenna Shang, Xiaofeng Jiang, Zhengbing Lyu, Jianqing Chen, Guodong Chen

**Affiliations:** 1School of Life Sciences, Central South University, Changsha 410031, China; 2023210901066@mails.zstu.edu.cn (W.T.); 2023210901110@mails.zstu.edu.cn (K.Z.); 223105@csu.edu.cn (S.S.); 202220901107@mails.zstu.edu.cn (B.Z.); 2College of Life Sciences and Medicine, Zhejiang Provincial Key Laboratory of Silkworm Bioreactor and Biomedicine, Zhejiang Sci-Tech University, Hangzhou 310018, China; 202230903190@mails.zstu.edu.cn (Z.L.); 202220901083@mails.zstu.edu.cn (J.Y.); 202220901027@mails.zstu.edu.cn (L.J.); 2023220903031@mails.zstu.edu.cn (L.Y.); 2023210901058@mails.zstu.edu.cn (W.S.); xfjiang@zstu.edu.cn (X.J.); zhengbingl@zstu.edu.cn (Z.L.); cjqgqj@zstu.edu.cn (J.C.); 3Zhejiang Sci-Tech University Shaoxing Academy of Biomedicine Co., Ltd., Shaoxing 312369, China

**Keywords:** IgNAR, VNAR, nurse shark, nanobody

## Abstract

The variable domain of IgNAR shows great potential in biological medicine and therapy. IgNAR has been discovered in sharks and rays, with the nurse shark (*Ginglymostoma cirratum*) IgNARs being the most extensively studied among sharks. Despite being identified in nurse sharks over 30 years ago, the characteristics and genomic localization of IgNAR remain poorly defined, with significant gaps even in the latest released genome data. In our research, we localized the IgNAR loci in the nurse shark genome and resolved the previously missing regions. We identified three IgNAR loci, designated *GcIgNAR1*, *GcIgNAR2*, and *GcIgNAR3*, with only *GcIgNAR1* and *GcIgNAR2* being expressed. GcIgNAR1 and GcIgNAR2 belong to type I and type II IgNARs, respectively, and each exhibits several different isoforms. Most nurse shark IgNARs possess five constant domains. However, we found transcripts of GcIgNAR1 and GcIgNAR2 lacking two constant domains, C4 and C5, which differ from the IgNAR of the whitespotted bamboo shark. The protein structures of GcIgNAR1 and GcIgNAR2, generated by AlphaFold3, confirmed the accuracy of the IgNAR loci we identified. Our findings advance scientific understanding of IgNAR in nurse sharks and facilitate future research and medical applications.

## 1. Introduction

Antibodies, also known as immunoglobulins (Ig), are critical components of the adaptive immune system, playing a vital role in identifying and neutralizing pathogens such as bacteria, viruses, and toxins [1,2,3]. These Y-shaped proteins are produced by B cells and are highly specific to antigens, which are molecules recognized as foreign by the immune system [3,4]. The specificity and diversity of antibodies are attributed to their unique variable regions, which undergo somatic recombination, hypermutation, and class switching to generate a vast repertoire capable of recognizing an extensive array of antigens [5,6,7]. There are five main classes of antibodies (IgM, IgD, IgG, IgA, and IgE) found in mammals; however, it is important to note that other isotypes exist in different species outside of mammals [8,9,10].

Heavy chain antibodies (HcAbs) and new antigen receptors (IgNARs) represent intriguing deviations from conventional antibody structures, offering unique insights and applications in immunology and biotechnology [11,12,13]. Unlike traditional antibodies, which are composed of both heavy and light chains, HcAbs and IgNARs consist solely of heavy chains (Figure 1A), resulting in distinctive structural and functional properties [13,14]. HcAbs were first discovered in camelids and are characterized by the absence of light chains [15]. This unique feature results in a simplified structure, where the antigen-binding site is formed exclusively by the variable region of the heavy chain (VHH), commonly known as a nanobody [16,17]. Nanobodies are remarkably small (approximately 12–15 kDa), highly stable, and capable of binding antigens with high affinity and specificity [18,19,20]. Their small size and robustness allow them to access epitopes that are often inaccessible to conventional antibodies, making them valuable tools in therapeutic and diagnostic applications [21]. Similarly, IgNARs were discovered in sharks and rays [22]. IgNARs also lack light chains and possess a single variable domain (VNAR) that binds antigens [23,24,25]. The structural simplicity and stability of VNARs make them analogous to camelid nanobodies, offering comparable advantages in terms of antigen-binding capabilities and potential applications [14].

The unique structural features of VNAR make it a promising candidate for therapeutic applications in various fields of medicine. Due to their small size and stability, VNARs can effectively bind to challenging epitopes, including those on proteins that are less accessible to conventional antibodies. For instance, VNARs have been developed as potential therapeutic agents against cancer [26,27,28], where they can be engineered to target specific tumor antigens, enhancing the precision of cancer treatments. Additionally, recent studies have highlighted their efficacy in neutralizing viruses, such as SARS-CoV-2 [29], showcasing their potential in infectious disease management. The versatility of VNARs extends to diagnostic applications as well [30], where they can be utilized in biosensors for detecting pathogens or disease markers.

Utilizing the latest genomic and transcriptomic data, we identified three IgNAR loci (*GcIgNAR1*, *GcIgNAR2*, and *GcIgNAR3*) in the nurse shark genome and successfully resolved previously missing regions. Furthermore, we analyzed the expression levels and isoforms of these loci, providing valuable insights into their functional diversity and potential roles in immune responses. Our findings offer critical insights into the diverse isoforms and structural characteristics of IgNARs in nurse sharks, thereby enhancing our understanding of their immune functions and evolutionary significance. This research not only advances the characterization of IgNARs in nurse sharks but also establishes a foundation for future studies exploring their potential applications in biological medicine and therapy.

## 2. Results

### 2.1. Three IgNAR Loci Discovered in Nurse Shark Genome Scaffold JAHRHZ010000107.1, with Only Two Expressed

IgNARs were identified in the nurse shark (*Ginglymostoma cirratum*) approximately 30 years ago (Figure 1B). However, the gene loci of IgNARs in the nurse shark genome have not yet been well defined. To identify the IgNAR gene loci in the nurse shark genome, we used the reported nurse shark IgNAR sequence (NCBI accession: AAB48195.1) as a query and employed the tblastn tool to search the latest released nurse shark genome (GCA_024137785.1 ASM2413778v1). We identified three IgNAR loci in the nurse shark genome, designated as *GcIgNAR1*, *GcIgNAR2*, and *GcIgNAR3* (Figure 1C). All three loci are clustered together on scaffold JAHRHZ010000107.1. Unfortunately, all three IgNAR loci contained missing regions (Figure 1C, red lines). To address this, we realigned the sequences using the published data (PRJNA732133, 60coverage) and successfully resolved the missing regions for *GcIgNAR1* and *GcIgNAR2* (Appendix A).

To determine the expression levels of *GcIgNAR1*, *GcIgNAR2*, and *GcIgNAR3*, we analyzed RNA-seq data (PRJNA841433 and GSE232302). We found that the expression levels of *GcIgNAR1* and *GcIgNAR2* can reach as high as 1500 FPKM, with both loci exhibiting similar expression levels, consistent with previous reports [31]. In contrast, the expression level of *GcIgNAR3* was too low to be detected (Figure 1D,E). These data suggest that *GcIgNAR1* and *GcIgNAR2* are the active IgNAR gene loci.

### 2.2. GcIgNAR1 and GcIgNAR2 Transcripts Respectively Exhibit at Least Eleven and Four Different Isoforms

RNA splicing happens during gene expression to generate different isoforms commonly; we wanted to know whether *GcIgNAR1* and *GcIgNAR2* could generate different isoforms. To address this question, we analyzed the exon–intron structures of *GcIgNAR1* and *GcIgNAR2* based on RNA-seq data. Given that splicing frequently occurs in the variable region, we could only delineate the borders of the variable domain coding region (VNAR). Consequently, we treated the entire variable coding region as a single exon. Our analysis revealed that *GcIgNAR1* contains ten exons, while *GcIgNAR2* has nine exons. The variable region was localized in exon 2 for both genes (Appendix A and Figure 2A,B). Apart from the variations in the variable region, we identified at least eleven isoforms of *GcIgNAR1* and four isoforms of *GcIgNAR2* resulting from alternative splicing of the exons (Figure 2A,B). Interestingly, we found that each constant domain is encoded by a single exon in both *GcIgNAR1* (exons 4 to 8) and *GcIgNAR2* (exons 3 to 7) (Figure 2A,B). Surprisingly, we discovered two short isoforms of IgNAR in nurse sharks that lack constant domains 4 and 5, consistent with Rumfelt et al.’s report [32], in contrast to the short form IgNAR of the whitespotted bamboo shark, which lacks constant domains 2 and 3 (Figure 2A,B) [33]. Our findings provide insights into the diverse isoforms of IgNAR genes in nurse sharks, which could have important implications for their immune function and evolution. Further investigation is needed to elucidate the functional significance of these isoforms and their potential roles in the nurse shark’s immune response.

The VNAR is composed of five constant framework regions (FR1, FR2, FR3a, FR3b, and FR4) and four hypervariable regions (CDR1, HV2, HV4, and CDR3). Notably, the FR1, CDR1, FR2, HV2, FR3a, HV4, and FR3b regions are encoded within the same continuous reading frame, resulting in a constant total amino acid count from FR1 to FR3b. Despite the conserved framework region structure, the amino acid sequences within the CDR1, HV2, and HV4 hypervariable regions undergo diversification during the generation of IgNAR. This sequence variation is likely generated through deamination of DNA or RNA in these regions. In contrast to the hypervariable CDR1, HV2, and HV4 regions, the CDR3 region is encoded by an extended DNA sequence. The sequence variations in CDR3 are produced through a combination of RNA splicing and deamination processes (Figure 2C), suggesting a more diverse CDR3 region. These findings highlight the distinct mechanisms employed by the VNAR to generate sequence diversity. While deamination plays a key role in diversifying the CDR1, HV2, and HV4 regions, both splicing and deamination contribute to the variability of the CDR3 region.

### 2.3. The Characteristics of GcIgNAR1 and GcIgNAR2 Isoforms

To elucidate the characteristics of GcIgNAR1 and GcIgNAR2 isoforms, we extracted the amino acid sequences of these isoforms. Since splicing occurs internally within the so-called exon 2, we extracted the longest open reading frame sequences from this exon. Subsequently, we performed sequence alignment using ClustalX2. We found the sequences are highly conserved in the framework regions of the VNAR and the entire constant domains.

In the VNAR, our analysis revealed common cysteine residues in the FR1 and FR3b regions of all isoforms (Figure 3A). Specifically, cysteine residues were found in the FR2 region of GcIgNAR1 isoforms and in the CDR1 region of GcIgNAR2 isoforms (Figure 3A, and these findings are consistent with previous reports [31,34,35,36]. Due to unclear sequences in CDR3 and FR4, we could not determine the detailed distribution of cysteine residues in these regions. In the constant domains, we identified two cysteines in C2–C5 and three cysteines in C1 (Figure 3A), suggesting the potential formation of intradomain disulfide bonds. Interestingly, we also found an extra cysteine in the C3–C4 linkers (Figure 3A), indicating the possibility of interdomain disulfide bonds, as IgNARs are typically dimeric in nature.

To construct VNAR libraries, we designed primer pairs targeting the VNAR (Figure 3A–C). The forward primers (GcVNAR1_F and GcVNAR2_F) were localized in the FR1 region, and the reverse primers (GcVNAR1_R and GcVNAR2_R) were localized at the beginning of the C1 domain. These primers can be used for phage display assays and next-generation sequencing library preparation, facilitating future studies of IgNARs in nurse sharks.

### 2.4. GcIgNAR1 and GcIgNAR2 Belong to Type I and Type II IgNAR, Respectively

Four types of IgNAR have been reported in sharks. Type I contains three disulfide bonds: FR1-FR3b, FR2-CDR3, and CDR3-FR4. Type II contains two disulfide bonds: FR1-FR3b and CDR1-CDR3 (Figure 4A). As mentioned earlier, the sequences in the CDR3 and FR4 regions are not clear, preventing us from confirming the types of GcIgNAR1 and GcIgNAR2 based solely on our initial analysis. To address this issue, we retrieved all full-length VNAR sequences available in the NCBI database. We identified 109 VNAR sequences from GcIgNAR1 and 66 VNAR sequences from GcIgNAR2. We then counted the cysteines in the CDR3 region of these VNARs. Our analysis revealed that 87 of the 109 GcIgNAR1 VNARs had two or more cysteines, with the majority (75 of 109) containing two cysteines. In contrast, 49 of the 66 GcIgNAR2 VNARs had one or more cysteines, with the majority (55 of 66) containing only one cysteine (Figure 4B and Appendix A). These data suggest that GcIgNAR1 and GcIgNAR2 belong to type I and type II IgNAR, respectively, consistent with the IgNARs identified by Diaz et al. [31].

Although the structures of certain domains of GcIgNARs have been reported [37], the full-length structures of GcIgNARs remain unknown. To further investigate the details of GcIgNAR1 and GcIgNAR2, we predicted the structures of the GcIgNAR1-7 homodimer and GcIgNAR2-1 homodimer using AlphaFold3. As expected, each chain of GcIgNAR1-7 and GcIgNAR2-1 consists of one VNAR and five constant domains. Interestingly, we identified one disulfide bond within each constant domain. Notably, the middle cysteine in C1 (Cys246 in GcIgNAR1-7 and Cys211 in GcIgNAR2-1) does not participate in disulfide bond formation (Figure 4C,D). Additionally, we observed that the cysteines in the C3-C4 linker (Cys475 in GcIgNAR1-7 and Cys459 in GcIgNAR2-1) form disulfide bonds between IgNAR chains (Figure 4C,D). In the VNAR, the CDR1, HV4, and CDR3 regions, which are responsible for binding antigens, are spatially close, and we clearly observed the disulfide bond formation between FR1 and FR3b (Figure 4E,F). These findings confirm that the IgNAR loci we discovered are indeed true IgNAR loci.

## 3. Discussion

The variable domain of IgNAR, commonly referred to as VNAR, represents a promising area of investigation due to its low molecular weight and potential for diverse biomedical applications [20]. For over 30 years, nurse sharks have been pivotal to IgNAR research [24,38]; however, the detailed characteristics and gene loci associated with IgNAR within the nurse shark genome have not been thoroughly defined. In our study, we successfully identified the precise *IgNAR* loci in the nurse shark and resolved previously missing genomic regions. This work provides a valuable reference for future studies on nurse shark IgNAR, including the construction of VNAR libraries.

While both the whitespotted bamboo shark and the nurse shark possess short form IgNAR, notable differences exist in their characteristics and generation mechanisms. Specifically, we identified transcripts in the nurse shark featuring three constant domains that lack the C4 and C5 domains. In contrast, the short form IgNAR of the whitespotted bamboo shark lacks the C2 and C3 domains [33,39]. Furthermore, in the whitespotted bamboo shark, the short form IgNAR is generated from the *IgNAR3* locus [33], whereas the short form IgNAR in nurse sharks arises from the same loci as the full-length IgNAR through RNA splicing. This variability within the *IgNAR* gene family suggests that different species may employ distinct mechanisms for generating antibody diversity, highlighting the need for further investigation into the evolutionary implications of these findings.

Despite these advancements, several questions remain regarding the IgNAR loci in nurse sharks. The latest genome assembly contains numerous gaps that may obscure the full extent of IgNAR diversity. Our previous work identified four active *IgNAR* loci in the whitespotted bamboo shark, indicating that the nurse shark may similarly possess a comparable number of undiscovered loci. Discrepancies between the IgNAR sequences identified in our study and those previously reported in the NCBI database, such as AAB48195.1 [40], AAB42384.1, AAB48198.1, AAB48200.1, AAB48201.1, and AAB48202.1, further suggest that additional *IgNAR* loci may still be unidentified, necessitating continued genomic exploration.

Understanding how nurse sharks select for the expression of IgNAR over conventional immunoglobulins such as IgM and IgW [32] remains an open question that warrants additional research. Furthermore, in our investigations of the whitespotted bamboo shark, we observed that expression levels of different *IgNAR* loci varied in response to distinct antigens [33]. The fundamental mechanisms governing these expression patterns require further exploration. Drawing parallels with research conducted in mammals, we hypothesize that epigenetic factors—such as DNA methylation, histone modification, noncoding RNA, and chromatin remodeling [41]—may play significant roles in the selection of immunoglobulin types and *IgNAR* loci. However, the specifics of these processes necessitate further investigation.

In summary, our research provides significant insights into the genomic and functional diversity of IgNARs in nurse sharks, substantially enhancing our understanding of these unique antibodies. By identifying three distinct loci and resolving previously missing genomic regions, we elucidate RNA splicing mechanisms that produce diverse isoforms, including short-form IgNAR transcripts that differ from those in related species. We believe that our findings will accelerate the exploration of IgNAR-derived nanobodies and their potential therapeutic applications.

## 4. Materials and Methods

### 4.1. Whole Genome Sequencing Data Analysis and Missing Regions Fixation

The reference nurse shark genome (GCA_024137785.1 ASM2413778v1) was downloaded from the National Center for Biotechnology Information (NCBI) database. The nurse shark genome index for bowtie2 alignment was built using bowtie2-build. Paired-end raw whole-genome sequencing data (PRJNA732133) were also downloaded from the NCBI database. Adapters were trimmed using trim-galore. The trimmed reads were then mapped to the nurse shark genome index using bowtie2 in paired-end mode. Reads mapped around gap regions were extracted using the R packages Rsamtools (Version 2.22.0) and GenomicRanges (Version 1.58.0), and the extracted paired-end reads were manually assembled to fix the missing gaps in SnapGene (Version 6.0.2).

### 4.2. RNA-Seq Data Analysis and Exon Extraction

The reference nurse shark genome (GCA_024137785.1 ASM2413778v1) was downloaded from the NCBI database. The original scaffold JAHRHZ010000107.1 was replaced with our fixed scaffold JAHRHZ010000107.1 sequence. The nurse shark genome index for hisat2 alignment was built using hisat2-build. Raw RNA-seq data (PRJNA841433 and GSE232302) were downloaded from the NCBI Gene Expression Omnibus database. Adapters were removed using trim-galore, and the cleaned reads were aligned to the previously generated genome index using HISAT2 (Version 2.2.0) in paired-end mode to produce BAM files. Exon sequences were manually extracted in the Integrative Genomics Viewer (IGV) using the generated BAM files. IgNAR isoforms, generated through RNA splicing, were also identified from the BAM files within IGV.

### 4.3. Protein Structure Prediction Using AlphaFold3

Protein structures for GcIgNAR1-7 and GcIgNAR2-1 were predicted using the AlphaFold artificial intelligence system. The amino acid sequences for these proteins were submitted to the public AlphaFold web server (https://alphafoldserver.com/) operated by Google DeepMind (accessed on 25 July 2024). Given that IgNARs typically form homodimers, we also predicted the protein structures of GcIgNAR1-7 and GcIgNAR2-1 in their homodimeric forms. The top predicted structure with the lowest predicted local distance difference test (pLDDT) scores, indicating the highest confidence in the structural prediction, was selected for further analysis. PyMOL (Version 2.5, Schrödinger, LLC, New York, NY, USA) was used to visualize and analyze the predicted protein structures.

## Figures and Tables

**Figure 1 ijms-25-12879-f001:**
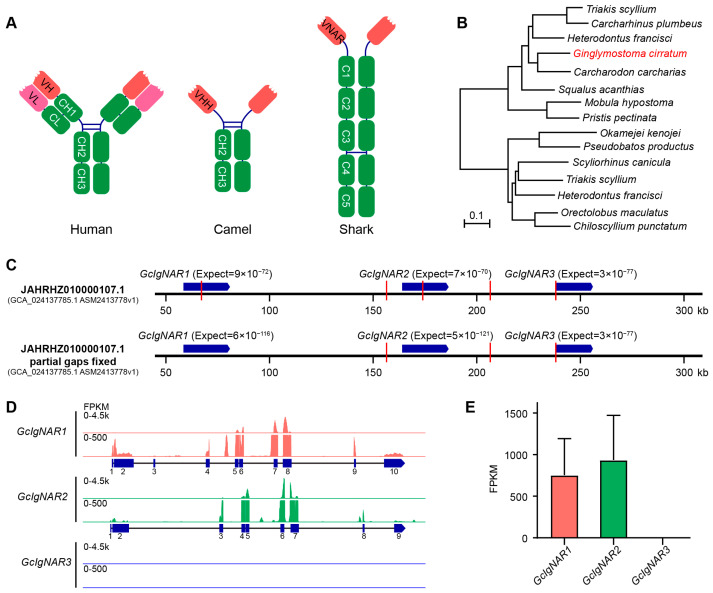
Genomic organization and expression of IgNAR in the nurse shark (*Ginglymostoma cirratum*). (**A**) Schematic representations of the characteristic antibody structures found in humans (IgG), camels (HcAb), and sharks (IgNAR). Human IgG is a tetrameric molecule, while camel HcAb and shark IgNAR are homodimeric. (**B**) Evolutionary tree depicting the phylogenetic relationships of IgNAR across various shark and ray species. The nurse shark (*Ginglymostoma cirratum*) is highlighted in red. (**C**) The genomic organization of the three IgNAR loci (*GcIgNAR1*, *GcIgNAR2*, and *GcIgNAR3*) identified in the nurse shark genome. The IgNAR loci are shown in blue, and regions with missing sequence information are indicated in red. The missing regions in *GcIgNAR1* and *GcIgNAR2* have been resolved. (**D**) RNA-seq data tracks illustrating the expression patterns of the three nurse shark IgNAR loci (*GcIgNAR1*, *GcIgNAR2*, and *GcIgNAR3*). The exon structures are depicted below the tracks. (**E**) The relative expression levels of the three nurse shark IgNAR loci (*GcIgNAR1*, *GcIgNAR2*, and *GcIgNAR3*).

**Figure 2 ijms-25-12879-f002:**
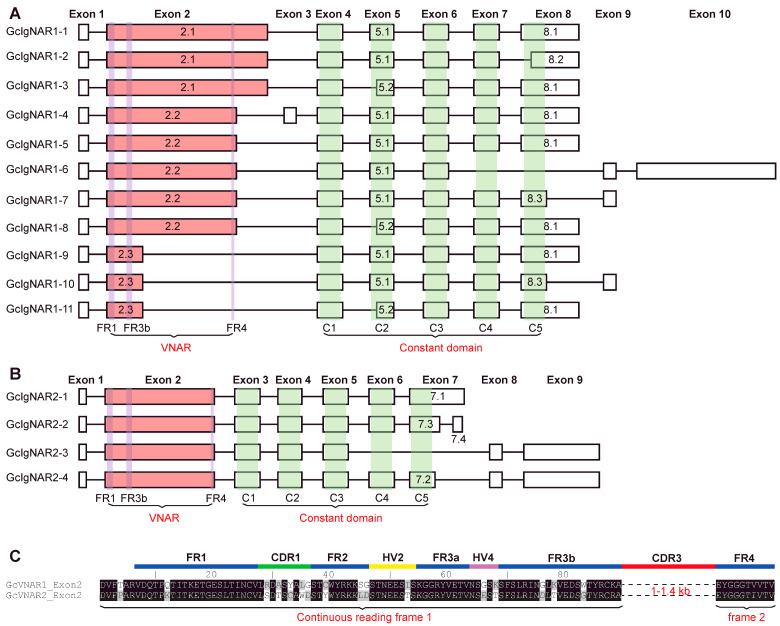
Isoform diversity of *GcIgNAR1* and *GcIgNAR2*. (**A**,**B**) The exon splicing patterns of *GcIgNAR1* (**A**) and *GcIgNAR2* (**B**). *GcIgNAR1* has 11 isoforms, while *GcIgNAR2* has 4 isoforms. Most of the *GcIgNAR1* and *GcIgNAR2* isoforms contain five constant domains, except for GcIgNAR1–6 and GcIgNAR2–3, which have lost constant domains 4 and 5. This is in contrast to the short form of IgNAR found in the whitespotted bamboo shark, which lacks constant domains 2 and 3. The framework regions FR1, FR3b, and FR4 are highlighted in purple, and the constant domains are highlighted in green. Exon numbers are indicated at the top of the figure. Each rectangle represents an individual exon, with exon variant numbers labeled within the rectangles. (**C**) The amino acid sequences of exon 2 for *GcIgNAR1* and *GcIgNAR2*. The continuous reading frame from FR1 to FR3b suggests that the amino acid number of these framework regions is maintained during the generation of IgNAR isoforms. The mutations in the CDR1, HV2, and HV4 may be introduced through deamination in DNA or RNA levels.

**Figure 3 ijms-25-12879-f003:**
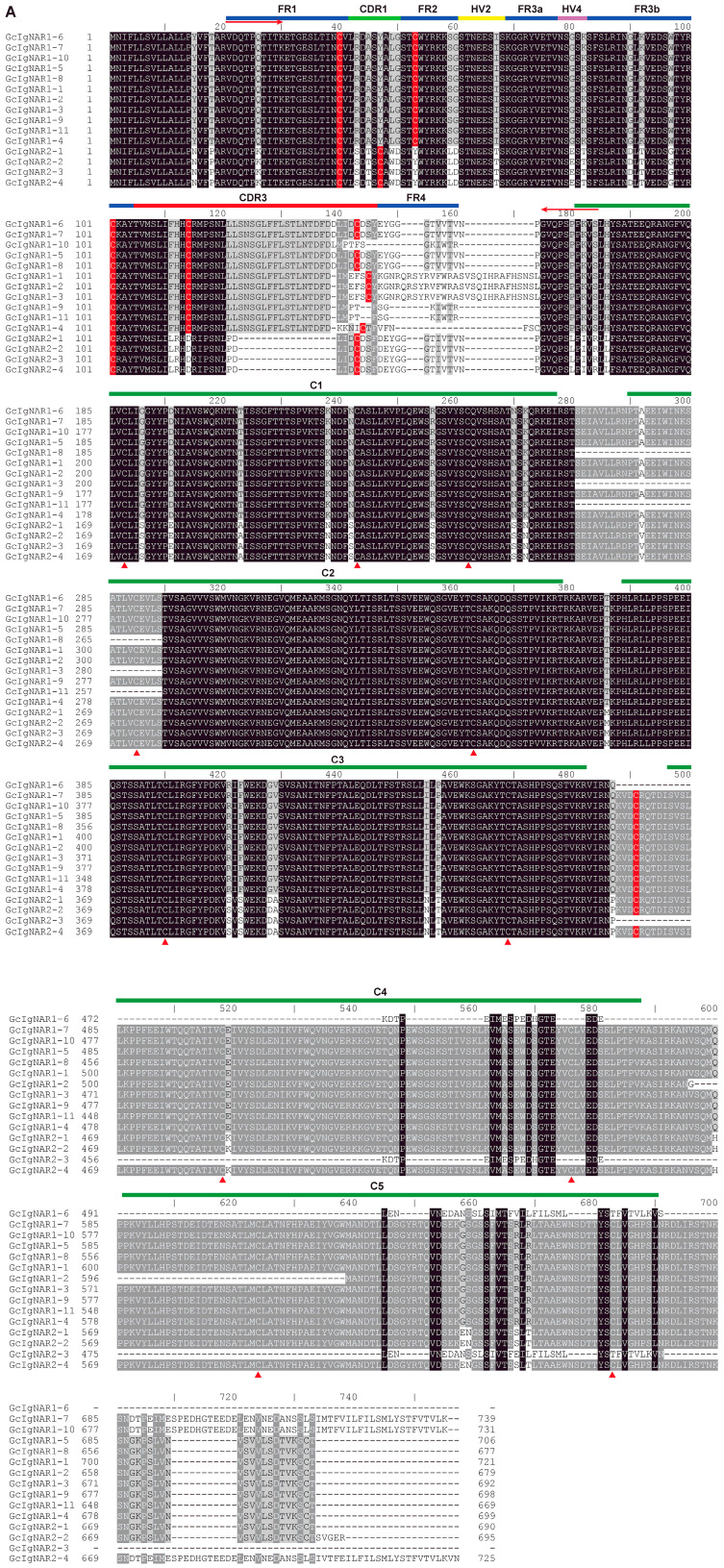
Characteristics of the GcIgNAR1 and GcIgNAR2 isoforms. (**A**) The amino acid sequences of the GcIgNAR1 and GcIgNAR2 isoforms. The domains are indicated at the top of the sequences. Cysteine residues in the VNARs are highlighted in red, and cysteine residues in the constant domains are marked with red triangles. The cysteine residues in the C3–C4 linkers are also highlighted in red. The binding sites for the primers used for VNAR amplification are indicated with red arrows in the FR1 region and at the beginning of the C1 domain. (**B**,**C**) The sequences (top) and binding sites (highlighted in red) of the forward (**B**) and reverse (**C**) primers used for VNAR amplification. The red letters in the primer sequences indicate the nucleic acid differences between GcIgNAR1 and GcIgNAR2. Similarly, the red letters in the amino acid sequences highlight the differences in amino acids between GcIgNAR1 and GcIgNAR2.

**Figure 4 ijms-25-12879-f004:**
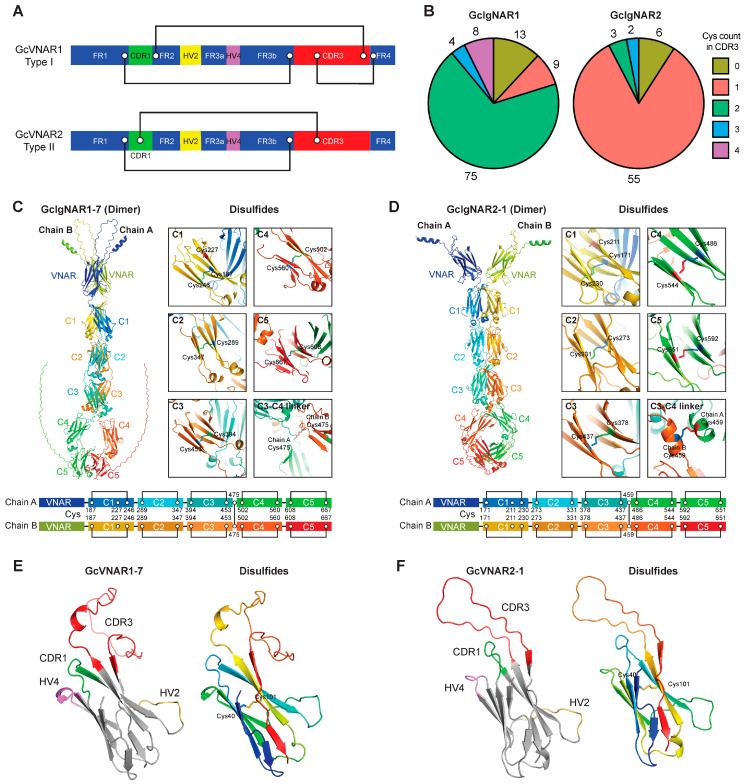
Structures of GcIgNAR1 and GcIgNAR2. (**A**) Model of type I and type II VNAR structures. Disulfide bonds are shown as black lines, and cysteine residues are labeled with white dots. Based on the positions of the cysteine residues, GcIgNAR1 belongs to the type I VNAR class, while GcIgNAR2 belongs to the type II VNAR class. (**B**) Pie charts showing the cysteine count in the CDR3 region of reported IgNARs or VNARs from the nurse shark. A total of 109 IgNARs or VNARs are from the GcIgNAR1 locus, and 66 IgNARs or VNARs are from the GcIgNAR2 locus. (**C**,**D**) Predicted protein structures of GcIgNAR1-7 (**C**) and GcIgNAR2-1 (**D**) using AlphaFold3. Disulfide bonds and cysteine residues are highlighted and labeled. The VNAR and constant domains are also labeled. Cartoon models are shown below the structures, with disulfide bonds depicted as black lines. (**E**,**F**) The VNAR structures of GcIgNAR1-7 (**E**) and GcIgNAR2-1 (**F**). The CDR1, HV4, and CDR3 regions are spatially close, and the disulfide bonds and cysteine residues between FR1 and FR3b are highlighted and labeled.

## Data Availability

The data that support the findings of this study are available from the corresponding author upon reasonable request.

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
