# Peer review of "Characteristics and Genomic Localization of Nurse Shark (Ginglymostoma cirratum) IgNAR"

_ijms, 2024, doi:10.3390/ijms252312879_

Round 1

Reviewer 1 Report

Comments and Suggestions for Authors

Comments for authors
The Tang et al characterized and localized IgNAR in Nurse Shark genome. Their findings help other researchers to study more about IgNAR and their applications. Eventhough, the manuscript has been written well and supported with figures, there is some minor corrections that have to be rectified before going for publication.
The introduction part and 3.1 section first paragraph looks similar to each other. If you see the context both are dealing the same things. It has to be changed. For instance, the nanobodies and their molecular weights were already given in introduction but it was repeated in the section 3.1.
Figure 2: Authors need to indicate what the numbers in the exons indicate?
Figure 3: Please indicate what the red letters in B-C indicate?
Line 276: Check the spelling of “several”.
Line 283: Suitable references need to be cited. For instance in this line, it was indicated that “some reported sequences” but the references for this reported sequences have not been mentioned. So authors need to provide suitable references.
It would also be better if you include some of the studies that have been done using IgNAR for any therapeutic applications either in discussion or in introduction that would help the readers to understand the use and importance of IgNAR.

Comments on the Quality of English Language

English language is fine

Reviewer 2 Report

Comments and Suggestions for Authors

The authors have reconstructed the IgNAR gene sequence using the public genome database of the nurse shark, and have made observations regarding gene region structure and the 3D structure. Below are the concerns I have with this manuscript.

1. Overall, I find that the novelty of this paper is insufficient. The authors reanalyze the IgNAR gene region using existing public databases, but this alone seems insufficient to justify a standalone paper. It is also unclear whether the authors' research is aimed at clinical applications or at advancing our understanding of the antibody production mechanisms in cartilaginous fish. They particularly emphasize the usefulness of IgNAR as a nanobody, but clinical applications of nanobody are already being pursued by other research groups, and it appears that the current databases are adequate in that regard.

2. In line 17, the authors state that IgNAR has been found in camels, sharks, and rays, but as mentioned later in the manuscript, the HcAbs in camels is not IgNAR. Such an abstract could significantly undermine the trust of readers and reviewers.

3. In lines 104–113 and the following content, the authors discuss material that belongs in the Introduction part. In the Results section, the findings should be presented concisely, and any redundant content should be avoided. 

4. In line 293, the isotypes of antibodies produced by cartilaginous fish should be verified using the appropriate references.

Round 2

Reviewer 1 Report

Comments and Suggestions for Authors

The authors have revised the manuscript well based on the comments. The revision improves the clarity of the manuscript including figures. Now, the manuscript can be accepted for publication.

Author Response

It is a great honor to receive your recognition of this work.

Reviewer 2 Report

Comments and Suggestions for Authors

I have read the cover letter with appreciation. The letter conveyed a strong passion for this research, allowing me to better understand the significance of the study and the value of its findings.

However, such passionate content should ideally be included not in a letter to the reviewer but within the manuscript’s Discussion section to readers. Unfortunately, when I read the first manuscript, I was unable to grasp this level of significance in the study. If the authors are capable of articulating the research’s importance as they did in the letter, this should be reflected within the manuscript itself. Therefore, I believe that the Discussion section should be rewritten.

In addition, the content from lines 116 to 124 in the Results section repeats what is already stated in the Introduction, making it feel unnecessarily redundant and diminishing the quality of the manuscript. I suggest starting directly from the content in line 125. (A minor comment in line 117, I do not believe that the discovery of the HcAbs and IgNARs is “recently”.)

In line 253, is there any evidence that nurse sharks express IgD and IgE?

The styles of references 3, 5, 15, 23 and 38 should be checked.

Round 3

Reviewer 2 Report

Comments and Suggestions for Authors

I would like to express my respect for the authors’ efforts in revising the manuscript. The improvements made throughout the revision process are evident, and I am pleased to recommend the manuscript for acceptance. Now I only have a small concern; in the reference 15, it does not have a journal title or that kind of book title?